# Sparse Symplectically Integrated Neural Networks

**Daniel M. DiPietro, Shiying Xiong, Bo Zhu**
Dartmouth College, Department of Computer Science
`{daniel.m.dipietro.22, shiying.xiong, bo.zhu}@dartmouth.edu`

## Abstract

We introduce Sparse Symplectically Integrated Neural Networks (SSINNs), a novel model for learning Hamiltonian dynamical systems from data. SSINNs combine fourth-order symplectic integration with a learned parameterization of the Hamiltonian obtained using sparse regression through a mathematically elegant function space. This allows for interpretable models that incorporate symplectic inductive biases and have low memory requirements. We evaluate SSINNs on four classical Hamiltonian dynamical problems: the Hénon-Heiles system, nonlinearly coupled oscillators, a multi-particle mass-spring system, and a pendulum system. Our results demonstrate promise in both system prediction and conservation of energy, often outperforming the current state-of-the-art black-box prediction techniques by an order of magnitude. Further, SSINNs successfully converge to true governing equations from highly limited and noisy data, demonstrating potential applicability in the discovery of new physical governing equations.

## 1 Introduction

Neural networks have demonstrated great ability in tasks ranging from text generation to image classification [33, 9, 8, 49, 20]. While impressive, most of these tasks can be easily performed by an intelligent human. The novelty is in a machine performing the task, rather than the task itself. Researchers are becoming increasingly interested in the role that machine learning can play in assisting humans–a role of discovery rather than of automation [16, 34, 21]. To this end, there has been significant research interest in applying machine learning to physical dynamical systems [12, 6, 24, 25, 32, 3, 35, 4]. Physical dynamical systems are everywhere, from weather to astronomy to protein folding [26, 15, 38, 44]. The underlying dynamics of many of these systems have yet to be unraveled, and many evade accurate prediction over time. Building accurate predictive models can spur significant technological, medical, and scientific advancement. Unfortunately, data from physical systems is often challenging to acquire and tainted with measurement and discretization error. Consequently, a primary challenge in this domain is that data-driven techniques must be able to cope with highly limited and noisy data.

Our work seeks to answer the following question: given the historical data of some physical dynamical system, how can we not only predict its future states, but also discern its underlying governing equations? We focus specifically on energy-preserving systems that can be described in the Hamiltonian formalism. There is a growing body of research in this domain, but much of it relies upon black-box machine learning techniques that solve the prediction problem while neglecting the governing equations problem [12, 6]. Namely, there are two primary paradigms for incorporating physical priors into neural networks: by embedding structure-enforcing algorithms into the network architecture, (e.g., a symplectic integrator [6]), or by enforcing a set of known mathematical constraints (e.g., the Navier-Stokes equations) in the loss function [36]. Integrator-embedded networks function as a black-box, and networks that use modified loss functions assume some knowledge of the system beforehand. For machine learning to truly play a role in human discovery, interpretability should be a paramount consideration.

Sparse regression has been used with great success to discover the underlying mathematical formulas of dynamical systems [4, 28, 48]. In this paper, we combine successful ideas from sparse regression equation discovery and black-box prediction techniques to introduce a new model, which we call Sparse Symplectically Integrated Neural Networks (SSINNs). Like many previous approaches, SSINNs ultimately seek to parameterize an equation called the Hamiltonian, from which a system can be readily predicted. To assist in this parameterization, our models incorporate a fourth-order symplectic integrator to ensure that the symplectic structure of the Hamiltonian is preserved, an embedded physics prior only employed in black-box approaches thus far. This also allows for continuous-time prediction. To incorporate interpretability, we utilize sparse regression through a mathematically elegant space of functions, allowing our models to learn which terms in the function space are part of the governing equation and which are not. This sparsity prior holds for nearly all governing equations and, experimentally, improves prediction performance and equation convergence.

Constructing a model in this way comes with a number of benefits over both black-box prediction techniques and current state-of-the-art methods for learning underlying equations. Black-box methods operate within incredibly large function spaces and, as a result, frequently converge to complicated functions that approximate the underlying dynamics of the system but offer no insight into its true governing equations. Unlike black-box methods, SSINNs are interpretable; by this, we mean that, once trained, a mathematically elegant governing equation that oftentimes represents the true governing equation of the system can easily be extracted from the model. Due to this interpretability, SSINNs also maintain far fewer trainable parameters (often <1% of black-box models) and consequently do not require specialized hardware to use once trained. When compared to current methods for learning governing equations, SSINNs incorporate a symplectic bias that reduces the number of possible solutions by placing restrictions on the numerical integration of the learned equation.

In summary, we propose SSINNs, which make the following contributions[1]:

- Incorporate symplectic biases to augment learning governing equations from data.
- Outperform state-of-the-art black-box prediction approaches by an order of magnitude on multiple classical Hamiltonian dynamical systems.
- Succeed in learning nonlinear Hamiltonian equations from as few as 200 noisy data points.

## 2   Related Work

**Symplectic neural networks**   Greydanus et al. introduced Hamiltonian Neural Networks (HNNs) to endow machine learning models for dynamical systems with better physical inductive biases [12]. Hamiltonian Neural Networks learn a parameterization of the Hamiltonian in the form of a deep neural network $\mathcal{H}_\theta$, allowing the model to maintain a conserved, energy-like quantity. This neural network is trained using position/momentum pairs and, rather than optimizing the output of the network directly, this approach optimizes its gradients, an approach explored in other works as well [50, 42, 31]. Most recently, Chen et al. introduced Symplectic Recurrent Neural Networks (SRNNs), which are recurrent HNNs that incorporate symplectic integrators for training and testing [6]. SRNNs parameterize the Hamiltonian using two separate neural networks, $T_\theta$ and $V_\theta$, and optimize their gradients by backpropagating through a simple leapfrog symplectic integration scheme. Symplectic Recurrent Neural Networks also incorporate a number of techniques to minimize the impact of noise, such as multi-step or "recurrent" training, as well as initial state optimization. These changes allow SRNNs to outperform HNNs on many tasks. Hamiltonian Neural Networks have been augmented to form other architectures as well, such as Hamiltonian Generative Networks [47].

**Sparse regression for governing equation discovery**   Brunton et al. introduced Sparse Identification of Nonlinear Dynamics (SINDy), which interprets dynamical system discovery as a symbolic regression problem [4]. The key observation for SINDy is that, for most dynamical systems, the underlying governing function is sparse in some larger space of functions. With this in mind, a library of function spaces is assembled, and the terms of the governing equation are learned through LASSO regression. Note that this approach is not limited to Hamiltonian systems; it was successfully employed to discover governing systems of partial differential equations. Since SINDy, many other works have explored similar symbolic regression approaches [28, 48]. Although separate from sparse

regression techniques, a number of other approaches have been proposed to couple accurate prediction and equation discovery, such as genetic algorithms, variational integrator networks, and various PDE-nets [7, 24, 25, 35, 40, 45].

**Predicting chaotic systems with machine learning**   Many Hamiltonian systems, including two examined in this work, exhibit chaotic motion. Chaotic dynamical systems are deterministic and characterized primarily by a sensitive dependence on initial conditions (SDIC). This implies that even initial conditions that are very close, albeit slightly different, will diverge exponentially in time [43]. A great deal of widely applicable dynamical systems are chaotic, such as weather, fluid, and celestial systems [26, 39, 38]. The accurate prediction of chaotic dynamical systems is an ongoing area of machine learning research [2, 27, 51, 1, 14, 30, 23, 19].

## 3   Framework

### 3.1   Hamiltonian Mechanics

William Rowland Hamilton introduced Hamiltonian mechanics in 1833 as an abstract reformulation of classical mechanics, similar in nature to the Lagrangian reformulation of 1788. Most physical systems can be described with the Hamiltonian formalism, and it is widely used in statistical mechanics, quantum mechanics, fluid simulation, and condensed matter physics [29, 37, 46, 41, 11, 13].

The state of a Hamiltonian system is described by variables $\mathbf{q} = (q_1, \ldots, q_n)$ and $\mathbf{p} = (p_1, \ldots, p_n)$, where $q_i$ and $p_i$ describe the position and momentum respectively of object $i$ in the system. Central to Hamiltonian mechanics is the Hamiltonian, $\mathcal{H}(\mathbf{q}, \mathbf{p})$, which generally corresponds to the total energy of the system. A Hamiltonian is *separable* if it can be written as $\mathcal{H} = T(\mathbf{p}) + V(\mathbf{q})$, where $T$ is kinetic energy and $V$ is potential energy; in this work, we consider only separable Hamiltonians.

Hamilton's equations uniquely define the time evolution of a Hamiltonian system as

$$\frac{d\mathbf{p}}{dt} = -\frac{\partial \mathcal{H}}{\partial \mathbf{q}}, \frac{d\mathbf{q}}{dt} = \frac{\partial \mathcal{H}}{\partial \mathbf{p}} \tag{1}$$

Thus, given some initial state $\mathbf{q_0} = (q_{0,1}, \ldots, q_{0,n})$ and $\mathbf{p_0} = (p_{0,1}, \ldots, p_{0,n})$, the time evolution of a Hamiltonian system is readily computable from this system of first-order differential equations. Importantly, this time evolution is symplectomorphic, meaning that it conserves the volume form of the phase space and the symplectic two-form $d\mathbf{p} \wedge d\mathbf{q}$.

Symplectic integration is a numerical integration scheme commonly employed for the time evolution of separable Hamiltonian dynamical systems. Any numerical integration scheme that preserves the symplectic two-form $d\mathbf{p} \wedge d\mathbf{q}$ is said to be a symplectic integrator. Symplectic integration has the advantage of conserving a slightly perturbed version of the Hamiltonian. For this reason, symplectic integrators enjoy widespread use in many fields of physics. This work employs a highly accurate fourth-order symplectic integration scheme [10].

### 3.2   SSINN Architecture

SSINNs consist of two specially constructed neural networks, $V_{\theta_1}(\mathbf{q})$ and $T_{\theta_2}(\mathbf{p})$, which parameterize the potential and kinetic energy of the Hamiltonian respectively; this design choice reflects the separability requirement for symplectic integration. Conceptually, each network performs a sparse regression within a user-specified function search space, which may be broadly defined to include multi-variate polynomial terms of specified degree, trigonometric terms, and so on. Each network computes the terms of the function basis within the forward pass; it is essential that this transformation happens within the network so that we may automatically compute gradients with respect to $\mathbf{p}$ and $\mathbf{q}$ later on, rather than with respect to any pre-computed terms. These basis terms are then passed through a single fully-connected layer. This layer learns the necessary terms of the basis, meaning that the trainable parameters become the coefficients of each basis term and are learned linearly with respect to each term in the basis. The architecture may be modified as necessary depending on the desired function space; additional layers with bias may be necessary if parameterizing within trigonometric functions (Equation 9 in Section 7). However, for most problems considered within this paper, a single fully-connected layer is all that is necessary. The output of this fully-connected layer is then potential or kinetic energy.

Coupling these two networks with a symplectic integration scheme for state-prediction and training is straightforward. Within our fourth-order symplectic integration scheme, each time that gradients of the Hamiltonian are required, we propagate through the networks, automatically compute the necessary gradients, and send them to the symplectic integrator (Figure 1). Fourth-order symplectic integration often involves many iterative computations depending on the length of the time-step, so there are frequently multiple passes through each network before loss is computed; hence there is some level of recurrence at play. Once the next state has been calculated, we compute the L1-norm between the predicted next state and the actual next state. Since the Hamiltonian is presumed sparse in our search space of terms, we incorporate L1-regularization so that only essential terms persist; although we do not incorporate thresholding, it is a useful technique that can be helpful for bringing non-essential terms completely to zero. Where $f$ is defined as the output of our fourth-order symplectic integrator, we define loss as

$$\mathcal{L}_{SSINN} = \left| f\left( \frac{dV_{\theta_1}}{d\mathbf{q}}, \frac{dT_{\theta_2}}{d\mathbf{p}}, \mathbf{q}_0, \mathbf{p}_0 \right) - (\mathbf{q}_1, \mathbf{p}_1) \right| + \lambda(\|\theta_1\| + \|\theta_2\|) \tag{2}$$

**Advantages over black-box prediction**  The primary motivation behind using our specially constructed $V_{\theta_1}$ and $T_{\theta_2}$ networks as opposed to deep neural networks is interpretability. While black-box methods can achieve impressive prediction performance, they bring humans no closer to understanding why systems work the way they do. Beyond interpretability, the structure of SSINNs results in significantly fewer trainable parameters–large SSINNs may contain *thousands* of trainable parameters, whereas large SRNNs may contain millions. This leads to smaller memory requirements and means that SSINNs do not require specialized GPU hardware to use once trained as they extract short, mathematically elegant governing equations. Further, since the function space of SSINNs is heavily reduced when compared to deep neural networks, they are at much less risk of overfitting and tend to learn underlying dynamics far more effectively.

**Advantages over current methods for learning governing equations**  Governing equations obtained through SINDy have no guarantees on their properties when integrated or used for time evolution. Specifically, consider the problem of learning a Hamiltonian through SINDy. Although SINDy can learn a system of differential equations, there is no guarantee that they will convert to a separable Hamiltonian. To learn a Hamiltonian directly, SINDy would optimize the function solely on the basis of maintaining a conserved quantity across sub-

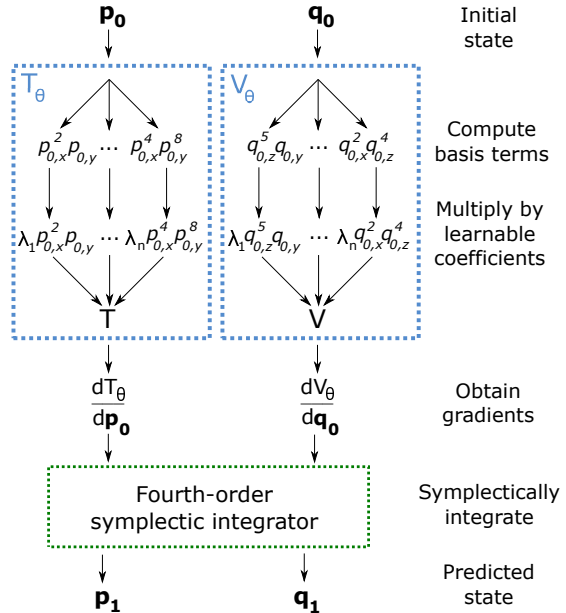

Figure 1: Using an SSINN to compute a predicted state. Note how terms are computed within the forward pass of each network, as well as the use of network gradients rather than direct network output. Although depicted as having a single linear layer where each trainable parameter is a coefficient, complicated function spaces may require several layers (for example, parameterizing within trigonometric functions).

sequent states. However, maintaining a conserved quantity does not guarantee that a Hamiltonian reflects the underlying dynamics of the system. For example, $\mathcal{H}(\mathbf{q}, \mathbf{p}) = 0$ maintains a conserved quantity for *every* physical dynamical system but has no valuable meaning for prediction. On the other hand, time evolution plays a crucial role in the optimization of equations obtained in SSINNs. Due to the incorporation of a symplectic integration scheme, equations learned through SSINNs are not only optimized to conserve symplectic structure, but also to accurately predict the system when numerically integrated. These added restrictions significantly reduce the number of possible solutions. Additionally, SINDy methods require estimating derivatives, which can degrade performance when the data is very noisy; as SSINNs integrate instead, they do not suffer from this downside. Other

discovery methods have leveraged Koopman theory, but these techniques require solving for the eigenvectors of the Koopman operator [17]. As SSINNs parameterize the Hamiltonian directly and do not require solving for these eigenvectors, they require relatively less data.

**Importance of the integrator**    We opt for a highly accurate fourth-order symplectic integrator. Less accurate integrators, especially those that are non-symplectic, introduce larger amounts of truncation error, which can substantially increase divergence in chaotic trajectories, even over small time-steps. Indeed, sparse networks trained with an RK4 integrator on the Hénon-Heiles system achieved an L1-error 4 orders of magnitude greater than their corresponding SSINNs, demonstrating that the fourth-order symplectic integrator is a major driver of performance. Additionally, the tolerance of the integrator can be adjusted to improve the speed of SSINNs; a tolerance of 0.01 is sufficient for convergence and results in training requiring no more than a few minutes.

## 4    Establishing a baseline: learning a simple chaotic Hamiltonian system

Our first experimental task seeks to rediscover the Hénon-Heiles governing Hamiltonian from data, as well as to use this Hamiltonian for system prediction over time. The Hénon-Heiles system was introduced in 1964 as an approximation of the plane-restricted motion of a star orbiting a galactic center [15]. It is defined as

$$\mathcal{H}(\mathbf{q}, \mathbf{p}) = \frac{1}{2}\left(p_x^2 + p_y^2 + q_x^2 + q_y^2\right) + q_x^2 q_y - \frac{1}{3}q_y^3 \tag{3}$$

The equipotential curves of this system form an interior triangular region that is inescapable when total energy is below $1/6$; motion is chaotic when total energy is above $1/12$ [43]. In this region, we have that $-1 < q_x < 1$ and $-0.5 < q_y < 1$.

### 4.1    Experimental setup and results

The Hénon-Heiles dataset consists of 5,000 points, each of which contains a $(\mathbf{q}, \mathbf{p})$ pair at $t = 0$ and $t = 0.1$. Initial states were randomly initialized within the interior triangular region with chaotic Hamiltonian values. We computed this dataset via Clean Numerical Simulation, a Taylor series scheme for approximating chaotic dynamical systems [22]. A noisy dataset was created by adding independent Gaussian noise ($\sigma = 0.005$) to all states in the initial dataset.

For both the noisy and clean dataset, we trained three SSINNs corresponding to 3rd, 6th, and 10th degree bivariate polynomial function spaces. Each model had an initial learning rate of $10^{-3}$ with decay and was trained using ADAM for 5 epochs on an RTX 2080 Ti system [18]. Hyperparameter tuning was minor, consisting of a small grid-search ($LR = [10^{-2}, 10^{-3}], L1 = [10^{-4}, 10^{-4}]$), sometimes followed by slight additional tweaking of regularization values. A regularization coefficient of $10^{-3}$ led to the best results. For comparison, we trained a 4-layer MLP with 400 nodes per layer as well as three SRNNs (1-layer, 2-layer, and 3-layer models); all SRNNs had 512 nodes

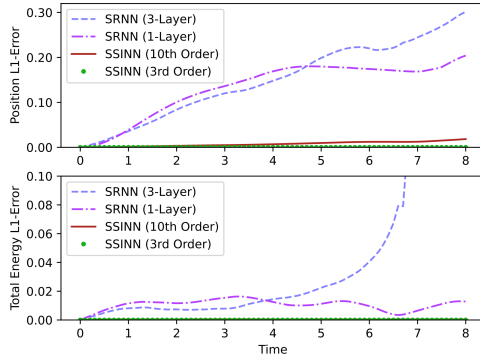

Figure 2: Mean L1-error when using models trained on the clean dataset to predict 50 validation Hénon-Heiles trajectories. Note that the significant increase in energy error for the 3-layer SRNN is due to a catastrophic failure on a single trajectory, demonstrating the risks of black-box models. Although not depicted, the performance of SINDY (3rd Order) lies between SSINN (3rd Order) and SSINN (10th Order) for both position and energy.

in each hidden layer. We replaced the original leapfrog integrator in the SRNNs with a fourth-order symplectic integrator to allow for fair comparison. Additionally, we used SINDy to learn a system of differential equations, constructed the Hamiltonian from the system, and unrolled its predictions with a symplectic integrator.

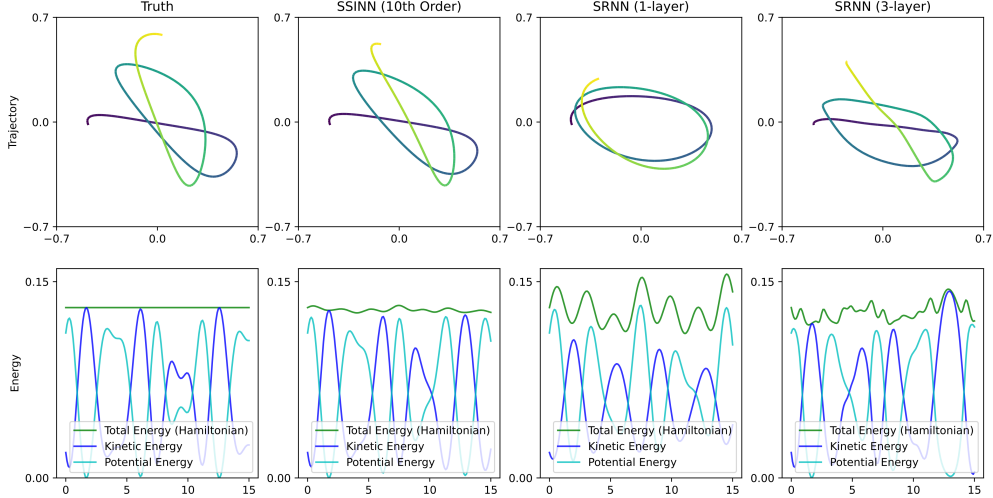

Figure 3: Predicting the Hénon-Heiles system with initial state $\mathbf{q_0} = (-0.48, -0.02)$, $\mathbf{p_0} = (-0.08, 0.18)$ from $t = 0$ to $t = 8$ using models trained on noisy data. All models were given a clean initial state to see how well they extracted true dynamics from noisy training. The SSINN predicts the trajectory nearly exactly and only deviates in the last few time steps. All models learned to conserve the Hamiltonian, but the energy perturbations for the SSINN model are very reduced.

All SSINNs trained on clean data converged to the true governing equation with no prior knowledge of the system; the 3rd Order SSINN learned the Hénon-Heiles Hamiltonian most accurately to $10^{-5}$ precision, even though only 18.8% of its trainable terms belonged to the true Hamiltonian. The least accurate SSINN outperformed the most accurate SRNN by an order of magnitude on average for predicting from $t = 0$ to $t = 0.1$ for 800 test Hénon-Heiles trajectories. This difference in performance rose to three orders of magnitude when comparing the most accurate SSINN to the most accurate SRNN. Due to the chaotic nature of this system, differences in prediction and energy conservation become even more apparent over time (Figure 2). The MLP failed to predict accurately and performed at least an order of magnitude worse than all other models. SINDy succeeded in learning a system of differential equations that could be converted to a separable Hamiltonian, but the coefficients of this Hamiltonian were only learned to $10^{-2}$ precision.

When trained on noisy data, SSINNs still converged to true governing equations, albeit with only $10^{-1}$ precision for each coefficient. Differences in performance remained apparent even with noisy data, with the prediction accuracy between SSINNs and SRNNs still differing by approximately an order of magnitude (Figure 3). When trained on noisy data, the Hamiltonian obtained by SINDy could not be converted to a separable Hamiltonian.

Note that the total energy line is not constant for the models in Figure 3, even though they employ a fourth-order symplectic integrator, which should maintain energy as a conserved quantity. Symplectic integrators do maintain conserved quantities, but these small oscillations commonly arise in practice due to truncation error. These oscillations are more apparent for parameterized Hamiltonians because trainable parameters frequently have much longer decimal components than the coefficients in true Hamiltonians–as a result, they are more strongly affected by truncation error.

## 5   Learning from highly limited and noisy data

This experimental task showcases the performance of SSINNs when data is highly limited and contaminated with noise. To do this, we rediscover a coupled oscillation Hamiltonian from data and use it for prediction. Coupled oscillation is a common physical phenomenon where the behavior of one oscillator in the system influences the behavior of other oscillators in the system. This section employs a one-dimension two-particle nonlinearly coupled system with the governing Hamiltonian

$$\mathcal{H}(\mathbf{q}, \mathbf{p}) = \frac{1}{2}\left(p_1^2 + p_2^2 + q_1^2 + q_2^2 + k(q_1 q_2)^2\right) \tag{4}$$

This system exhibits chaotic motion when the coupling constant $k$ is 1 [43].

## 5.1 Experimental setup and results

Using a fourth-order symplectic integrator, we first generated a clean dataset of 500 randomly sampled state transitions with $k = 1$ and positions and momenta between $-1$ and 1 from $t = 0$ to $t = 0.1$. We also generated a noisy ($\sigma = 0.005$) dataset of only 200 randomly sampled state transitions from $t = 0$ to $t = 0.3$ to allow for multi-step training. All other aspects of the experimental setup remained unchanged from the Hénon-Heiles task, with the exception that we increased the initial learning rate to $10^{-2}$ and trained for 60 epochs to account for the decrease in data. Additionally, the regularization coefficient was tuned to $8 \cdot 10^{-3}$.

All SSINNs converged to the governing Hamiltonian on the clean dataset, with the best-performing model achieving prediction error of $2 \cdot 10^{-8}$ for computing $t = 0$ to $t = 0.1$ on a validation dataset of 100 points. In comparison, the best-performing SRNN only managed to achieve prediction error of $3 \cdot 10^{-3}$ over the same time period, a 100,000x difference in performance. The MLP only achieved $9 \cdot 10^{-3}$ prediction error. The system of differential equations obtained via 3rd-order SINDy converted to a separable Hamiltonian and achieved $1.5 \cdot 10^{-4}$ prediction error; each coefficient was learned to $10^{-1}$ precision, whereas SSINN coefficients were learned to $10^{-6}$ precision.

Prediction error worsened significantly for all models when trained on the noisy dataset. However, differences in performance still remained clear (Table 1). Even from only 200 noisy data points, SSINNs still converged to the true governing equation, although the precision for each coefficient was reduced from the clean dataset. For example, the 3rd order SSINN learned

$$\hat{\mathcal{H}}(\mathbf{q}, \mathbf{p}) = 0.497p_1^2 + 0.501p_2^2 + 0.498q_1^2 + 0.498q_2^2 + 0.505(q_1q_2)^2 \tag{5}$$

as the governing equation. For this task, the MLP performed significantly worse than both SSINNs and SRNNs, likely due to the lack of data. SINDy failed to achieve a separable Hamiltonian when trained on noisy data.

Table 1: Predicting 100 validation coupled-oscillator systems from $t = 0$ to $t = 0.1$ using models trained on noisy data. All initial states were clean to see how well models learned underlying dynamics from noisy training. Table contains average error, as well as each model's number of trainable parameters.

| Name | # Params | Position L1-Error | Momentum L1-Error |
|---|---|---|---|
| SSINN (3rd order) | 32 | $6.6 \cdot 10^{-4}$ | $5.6 \cdot 10^{-4}$ |
| SSINN (6th order) | 98 | $8.9 \cdot 10^{-4}$ | $9.7 \cdot 10^{-4}$ |
| SSINN (10th order) | 242 | $3.8 \cdot 10^{-3}$ | $3.1 \cdot 10^{-3}$ |
| SRNN (1-layer) | 3072 | $1.5 \cdot 10^{-1}$ | $1.9 \cdot 10^{-1}$ |
| SRNN (2-layer) | 527360 | $2.8 \cdot 10^{-1}$ | $9.5 \cdot 10^{-1}$ |
| SRNN (3-layer) | 1051648 | $8.6 \cdot 10^{-2}$ | $1.0 \cdot 10^{1}$ |

## 6 Predicting systems with many degrees of freedom

Thus far, we have only considered dynamical systems with 1 or 2 particles. Here we attempt to learn a mass-spring system with 5 particles and 6 springs (effectively a system of 5 coupled harmonic oscillators). This system is governed by Hamiltonian $\mathcal{H}(\mathbf{q}, \mathbf{p}) = V(\mathbf{q}) + T(\mathbf{p})$ where

$$V(\mathbf{q}) = \frac{k_1}{2}q_1^2 + \frac{k_2}{2}(q_2 - q_1)^2 + \frac{k_3}{2}(q_3 - q_2)^2 + \frac{k_4}{2}(q_4 - q_3)^2 + \frac{k_5}{2}(q_5 - q_4)^2 + \frac{k_6}{2}(L - q_5)^2 \tag{6}$$

$$T(\mathbf{p}) = \frac{p_1^2}{2m_1} + \frac{p_2^2}{2m_2} + \frac{p_3^2}{2m_3} + \frac{p_4^2}{2m_4} + \frac{p_5^2}{2m_5} \tag{7}$$

### 6.1 Experimental setup and results

We generated a dataset of 800 consecutive position-momentum pairs with a step-size $\Delta t = 0.1$ via fourth-order symplectic integration. Each mass was randomly initialized between 1 and 5 and evenly spaced from 0 to 1. Spring constants were randomly initialized between 0.05 and 0.4, and initial momenta were randomly initialized between -0.1 and 0.1. As done previously, a noisy dataset was also generated ($\sigma = 0.005$).

Due to the larger state vector, we adjusted our SSINN models to 2nd, 4th, and 6th degree polynomial function spaces and altered the regularization parameter to $4 \cdot 10^{-4}$. Similarly, we increased the SRNN models to 1024 hidden nodes per layer. All models were trained for 30 epochs with an initial learning rate of $10^{-2}$.

None of the SSINNs learned the exact governing Hamiltonian, but they all correctly identified the necessary sparsity in the function space, discovering the highly predictive terms that belonged to the true Hamiltonian. Even without learning the exact governing equation, SSINNs displayed

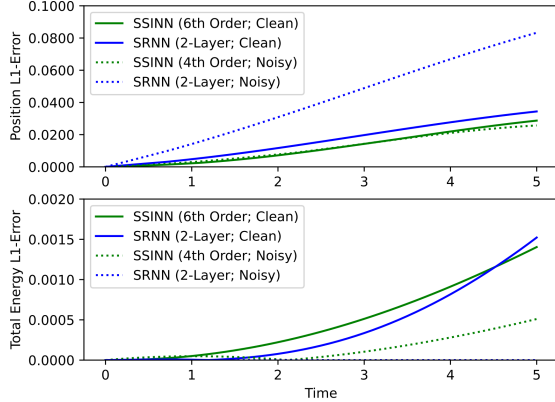

Figure 4: Predicting the mass-spring system 5 steps into the future using the best SSINNs and SRNNs from both clean and noisy training. Note that the SSINN actually had slightly decreased position error when trained on noisy data, whereas the error of the SRNN more than doubled.

highly accurate performance for both clean and noisy data. The performance of the best-performing SRNN and the best-performing SSINN was generally comparable, although SSINN models seemed less affected by noise (Figure 4). Notably, the best-performing SRNN has 1.05m trainable parameters, whereas the 6th order SSINN only has 390, a memory difference of approximately 2700 times despite performance being nearly identical. The system of differential equations discovered by SINDy could not be converted to a separable Hamiltonian.

## 7 Alternative function spaces

The preceding three dynamical systems could be exactly represented using only polynomial terms. Although many systems only have polynomial nonlinearities or are at least well-approximated by high degree polynomials, it is still important to demonstrate the robustness of SSINNs in alternative function spaces. To do this, we apply SSINNs to a simple pendulum system, which contains a trigonometric nonlinearity and is governed by the Hamiltonian

$$\mathcal{H}(q_\theta, p_\theta) = \frac{p_\theta{}^2}{2ml^2} - mgl\cos(q_\theta) + mgl \tag{8}$$

The constant term $mgl$ plays no role in the time evolution of the system and thus cannot be learned with SSINNs. For this task, we examined several function spaces. The first function space contained polynomial and trigonometric terms and was employed for learning both $V$ and $T$. It is defined with learnable parameters $\lambda_1, \ldots, \lambda_n$ as

$$f(x) = \lambda_1 x + \lambda_2 x^2 + \lambda_3 x^3 + \lambda_4 \sin(x + \lambda_5 x + \lambda_6) \tag{9}$$

Since the necessary cosine term is not included directly in the function space, the SSINN must shift the provided sine term by $\frac{\pi}{2}$ using parameter $\lambda_6$ to obtain a cosine term. Note that the interior of the sine term is structured so that the sparsity promotes a typical period of $2\pi$ rather than the period approaching infinity if $\lambda_5$ becomes very small; this was found to improve utilization of the trigonometric term. Parameterizing within the sine term in this manner results in a non-convex optimization, but this did not hinder performance in practice. For this task, we generated 200 steps of position-momentum data for a pendulum system with $g = l = 1$, $m = 2$, and initial state $q_\theta = 1.4$

and $p_\theta = 0$. All experiments were trained for 40 epochs with a regularization parameter of $10^{-4}$ and initial learning rate of $10^{-2}$. When both $V_\theta$ and $T_\theta$ were trained using the above function space, $T_\theta$ converged to the exact $T$. Interestingly, $V_\theta$ learned the first-order Taylor approximation of $V$ with a small corrective cosine term, which was accurate enough over the tested domain to provide $5.8 \cdot 10^{-6}$ prediction error. Such behavior could likely be minimized by shifting angular data by $2\pi$, but we wanted to avoid such manipulations that indicate an underlying knowledge of the system's structure.

Next, we restricted the function space so that $T_\theta$ possessed only third-degree polynomial terms and $V_\theta$ possessed only the trigonometric term included in Equation 9. With this function space, the SSINN converged to the true governing equation with $10^{-4}$ precision, correctly learning to shift the sine term in order to obtain a cosine term.

Thus far, all experiments have included bases that contain all necessary terms needed to learn the underlying Hamiltonian. What happens if this is not the case? To address this, we restricted the function space so that $T_\theta$ and $V_\theta$ possessed fifth degree polynomial terms but no trigonometric term. In this example, $T_\theta$ converged as usual to the correct governing $T$; $V_\theta$, on the other hand, converged to the fifth-order Taylor polynomial of $V$. Even though the trigonometric term was not included, this parameterized Hamiltonian achieved a noteworthy $7 \cdot 10^{-5}$ L1 prediction error per time step. These results indicate that SSINNs can successfully achieve highly precise prediction performance even if all necessary terms are not included in the basis. Obtaining the Taylor polynomial in this manner can potentially offer insight into the true underlying Hamiltonian, although we leave this as a topic for future work.

## 8 Scope and limitations

**User-defined function spaces**  As is the case for SINDy, SSINNs require a user-defined function space to extract terms from. This approach allows for control over the terms present in final governing equation and can even be used in an attempt to simplify known governing equations. That said, user-defined function spaces present challenges for less mathematically elegant systems, such as those with rational, exponential, or non-analytic terms. Even with this drawback, SSINNs offer a valuable first-step approach for learning dynamical systems before resorting to black-box techniques. Automatically augmenting the function space to accommodate system complexity or even incorporating evolutionary strategies may help solve this issue; evolutionary strategies have been used with great success already in the domain of physical dynamical systems [42]. We leave this as a topic for future work.

**Hamiltonian formalism**  Due to the incorporation of a symplectic integrator, SSINNs make a number of assumptions about the dynamical system that they are approximating, such as conservation of energy and separability. While these assumptions are generally useful physics priors, they can be limiting for real-world systems that lack conserved quantities. Including additional networks to better handle effects such as dampening may prove useful for better accommodating real-life systems. Similar techniques can also be employed to better handle noise.

**Lack of non-(p,q) data**  Several works have attempted to learn dynamics from data that is not already in $(\mathbf{p}, \mathbf{q})$ format, often incorporating autoencoders to accomplish this [12, 5]. Rather than focusing on these examples, we instead emphasize other common real-world limitations like small and noisy data. That said, $(\mathbf{p}, \mathbf{q})$ data is not always available, especially when working with real-life observatory data. We leave this as a topic for future work.

## 9 Conclusion

We introduced the Sparse Symplectically Integrated Neural Network (SSINN), a novel model for predicting and learning Hamiltonian dynamical systems from data. SSINNs incorporate a symplectic prior via the use of a symplectic integrator within the network; this assists in learning conservation of energy and allows for continuous-time prediction. SSINNs are also interpretable since they rely upon sparse regression through a mathematically elegant space of functions. Once SSINNs are trained, governing equations can be easily extracted and, more often than not, hand-written on a sheet of paper. SSINNs demonstrated impressive performance even when data was highly limited or noisy, frequently outperforming state-of-the-art black-box prediction techniques by significant margins while maintaining far fewer trainable parameters.

## Broader Impact

This research marks a significant advancement in using machine learning as a tool to assist in human discovery. We demonstrate that complex Hamiltonian physical systems–systems that in many cases have taken substantial human work and time to describe–can be readily learned and accurately predicted to very high precision from small amounts of data. As dynamical systems are everywhere, the impact of this work is far-reaching. Potential areas of noteworthy application include astronomy, medicine, engineering, and weather prediction. While our model itself does not necessarily bring about significant ethical considerations, researchers should take care whenever applying such models to domains prone to data bias, such as medicine.

## Acknowledgments and Disclosure of Funding

This project is support in part by the Dartmouth Neukom Institute CompX Faculty Grant, the Burke Research Initiation Award, and NSF MRI 1919647. Daniel M. DiPietro is supported in part by the Dartmouth Undergraduate Advising and Research Program (UGAR). We thank John McCambridge for valuable discussions. Finally, we are grateful for the insightful feedback provided by the anonymous NeurIPS reviewers.

## Footnotes

[1]Our code is available at `https://github.com/dandip/ssinn`

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
