[Supplementary Material]

# A   Supplementary Information for Fourth-Order Symplectic Integrator

Any numerical integration scheme that preserves the symplectic two-form $d\mathbf{p} \wedge d\mathbf{q}$ is said to be a symplectic integrator. These integrators are highly accurate and commonly employed for computing the time evolution of Hamiltonian systems. For a separable Hamiltonian $\mathcal{H} = T(\mathbf{p}) + V(\mathbf{q})$ with states $(\mathbf{q}_i, \mathbf{p}_i)$ at time $t_i$, a fourth order symplectic integrator may be implemented as shown in Algorithm 1. When implemented in PyTorch with tensor operations, we may use automatic differentiation to readily back-propagate through the integrator and accumulate losses.

---

**Algorithm 1:** Fourth-order Symplectic Integrator

---

**Input** : $\mathbf{q}_i, \mathbf{p}_i, t_i, t_{i+1}, \frac{\partial T}{\partial \mathbf{p}}, \frac{\partial V}{\partial \mathbf{q}}$, eps
**Output** : $\mathbf{q}_{i+1}, \mathbf{p}_{i+1}$

1 steps $\leftarrow (t_{i+1} - t_i)/(4 * \text{eps})$ ;
2 $h \leftarrow (t_{i+1} - t_i)/\text{steps}$ ;
3 $\mathbf{kq} \leftarrow \mathbf{q}_i$ ;
4 $\mathbf{kp} \leftarrow \mathbf{p}_i$ ;
5 **for** $i \leftarrow 1$ **to** *steps* **do**
6      **for** $j \leftarrow 1$ **to** 4 **do**
7         $\mathbf{tp} \leftarrow \mathbf{kp}$ ;
8         $\mathbf{tq} \leftarrow \mathbf{kq} + c_j * h * \frac{\partial T}{\partial \mathbf{p}}(\mathbf{kp})$ ;
9         $\mathbf{kp} \leftarrow \mathbf{tp} - d_j * h * \frac{\partial V}{\partial \mathbf{q}}(\mathbf{kq})$ ;
10       $\mathbf{kq} \leftarrow \mathbf{tq}$ ;
11     **end**
12 **end**
13 **return** $\mathbf{kq}, \mathbf{kp}$

---

Note that $c = [c_1, c_2, c_3, c_4]$ and $d = [d_1, d_2, d_3, d_4]$ are constants defined as follows:

$$c_1 = c_4 = \frac{1}{2(2 - 2^{1/3})} \tag{1}$$

$$c_2 = c_3 = \frac{1 - 2^{1/3}}{2(2 - 2^{1/3})} \tag{2}$$

$$d_1 = d_3 = \frac{1}{2 - 2^{1/3}} \tag{3}$$

$$d_2 = \frac{2^{1/3}}{2 - 2^{1/3}} \tag{4}$$

$$d_4 = 0 \tag{5}$$