[Reviews · NeurIPS 2020]

Review 1

Summary and Contributions: This paper introduces a new method for learning Hamiltonians from data. Rather than a black-box method, this approach results in learning actual symbolic equations. It's only the second paper I know of that attempts to learn simple Hamiltonian equations from data.

Strengths: I am quite excited about this paper. It's a significant step forward for learning Hamiltonians. - Black-box deep learning methods do not learn the equations. Additionally, the experiments in this paper against one previous paper show a significant improvement in accuracy of prediction, especially on long time scales. - It is non-trivial to apply SINDy-like methods to learn Hamiltonians, since you're dealing with a null-space issue (the "left side" is 0.) Additionally, SINDy generally struggles with amplifying noise due to needing to estimate derivatives of noisy data. This paper avoids this issue by integrating instead. I think this paper will be of interest to the NeurIPS community. The "Hamiltonian Neural Networks" paper by Greydanus, et al. at NeurIPS 2019 is quite popular with 59 citations on Google Scholar. Quite a few of those papers are building off of it already. Some other impressive details: - They are able to integrate accurately far past the interval of the training data. - The data is chaotic - The models are quite small

Weaknesses: There are some details left out that may make this hard to reproduce (see comments below). Do you intend to make the code public, or are you just sharing it with the reviewers? In addition to the below questions, I would suggest adding more details in a supplement about how the integration works. If I were to try to reproduce this paper, I think that would be a stumbling block.

Correctness: Please explain how you handled train/val/test data. I see reference in the paper to novel trajectories. In your code, I see "test" data being used as validation data. Were the "novel trajectories" completely held out & different from what was called "test" data in your code? Relatedly, I see that you checked off "The range of hyper-parameters considered, method to select the best hyper-parameter configuration, and specification of all hyper-parameters used to generate results." and "The exact number of training and evaluation runs." in the reproducibility checklist, but I don't see this in the paper. This is especially helpful to understand if the baselines were tuned a fair amount. The abstract should be edited to say that your method _sometimes_ outperforms state-of-the-art by an order of magnitude, since this wasn't consistently true (such as in Section 6.1). Also, I would like to see a stronger argument that comparing against the SRNN paper counts as "outperforming the current state-of-the-art black-box prediction techniques." Black-box prediction could be quite broad, but even if you restrict yourself to the methods that try to learn a Hamiltonian, is the SRNN paper the current state-of-the-art? There are quite a few papers in the last year using deep learning to learn Hamiltonians.

Clarity: Overall, I thought that the paper was very well-written and easy to read. I have some clarifying questions, though. How fast is your method for training and inference? Having so few trainable parameters probably helps, but I don't have a good sense of how slow the symplectic integrator might be. The whole neural network is just one fully-connected linear layer, right? Are you just learning the coefficient matrix, no bias? The paper would be easier to read if this was more explicit. When you say that the method converged to the "true governing equation," what do you mean there? The right sparsity pattern? Do you also have a threshold of how accurate the coefficients have to be? I might have missed this, but do you incorporate thresholding in addition to the L1 regularization? Usually this is done in SINDy to get the coefficients all the way to zero.

Relation to Prior Work: It would be good to note as a limitation (or future work) being able to handle data that's not already in (p, q) coordinates. The Hamiltonian NNs paper [10] has an example where the input is pixel data, not the coordinates (p, q). They are able to incorporate an autoencoder to learn good coordinates. Something similar has been done for SINDy with autoencoders in Champion, et al. "Data-driven discovery of coordinates and governing equations." PNAS 2019. In your paragraph " Advantages over current methods for learning governing equations," you should compare with this paper, which is one approach for using SINDy to discover conserved quantities: Kaiser, et al. "Discovering conservation laws from data for control" IEEE CDC 2018 It would be good to cite "Discovering Symbolic Models from Deep Learning with Inductive Biases" by Cranmer, et al. in your final paper, although you could not have been expected to cite it in this version. It was posted on arxiv two weeks after your paper was due. They learn symbolic Hamiltonians with a genetic algorithm approach.

Reproducibility: No

Additional Feedback: One aspect of your paper that is pretty cool and could be more emphasized is that, to my understanding, you avoid having to estimate derivatives of your data. This is a major downside of the original SINDy algorithm when data is noisy. Another is being able to parameterize inside the sine term in Equation 9. I believe most SINDy papers avoid this because it makes the optimization non-convex. UPDATE: Thank you for addressing my concerns in your response. I look forward to trying out your method once the paper is published.


Review 2

Summary and Contributions: This work addresses system identification for Hamiltonian systems. Instead of using a standard deep neural network (DNN) approach to approximate the Hamiltonian function H(q,p) (as explored in previous work), the Hamiltonian function is approximated via a chosen basis (similar to SINDY). Consequently, as the parameters to be estimated are the basis coefficients, instead of a much more general Hamiltonian function approximated via a DNN, the number of parameters is much smaller than for a DNN. A symplectic integrator is used to integrate the Hamiltonian equations. Overall, this is a nice idea with moderate technical novelty which shows promising results on some simple Hamiltonian toy problems. While the manuscript frequently mentions interpretability this has not been clearly demonstrated. Further, the benefits of using a symplectic integrator (while eminently sensible in the context of Hamiltonian equations) have not been explored, and previous work has tackled a bit more challenging setups by for example using images as inputs rather than assuming direct measurements of the position and momentum variable are available.

Strengths: 1. Sensible combination of symplectic integration, parameterization of the Hamiltonian via a chosen basis and identification of the Hamiltonian based on the estimation of basis coefficients. 2. Several toy experiments show improved results over the SRNN approach. 3. Guaranteed energy-preservation based on symplectic integration (though this does sometimes not appear to be the case in the figures, see below). 4. Efficient estimation of Hamiltonians by using a user-specified basis.

Weaknesses: 1. Other works (like SINDY or the Hamiltonian Neural Network work by Greydanus et al.) go beyond identification of systems with direct access to the position, q, and momentum variables, p. Instead they also explore identifying systems from images. It would have been nice to see such experiments to show the behavior of the approach when moving beyond the rather simple problems considered. 2. The paper strongly emphasizes the energy conservation properties of symplectic integrators. However, many of the plots show that the energy is not conserved, but may oscillate (Fig. 3) for SSINNs. Further. even though in the experiments the integrator for SRNNs is also symplectic the energy sometimes strongly changes. What causes this for SRNNs and SSINNs? 3. It remains unclear / unexplored what the most important driver for the good performance on the test problems is. Is it mostly the simplicity of the chosen basis (in particular, as it is chosen to contain the terms of the true Hamiltonians)? Or does the integrator indeed play a large role? Concretely, would the results get significantly worse when replacing the integrator by higher order Runge Kutta method, for example? (This will, of course, not preserve energy exactly, but might be good enough for a sufficiently small time-step; or are such integrators not applicable to the problems tested?) 4. It is unclear why the approach is not compared to SINDY as it uses a conceptually similar sparse modeling of the basis terms. 5. All the problems use bases that contain the terms of the exact Hamiltonian. What happens if this is no longer the case (e.g., when one wants to recover an a-priori unknown Hamiltonian)? Would this not create potential issues with the sparsity assumption? I.e., if a term is not included many of the included terms might need to compensate for it? 6. There is a strong emphasis on interpretability, but it is unclear why models with many polynomial terms are highly interpretable. A concrete example where better interpretability is achieved would have been useful. 7. Novelty is moderate as it is a combination of Hamiltonian networks (a la Greydanus et al.) with a variational integrator and a chosen basis (a la SINDY). However, it is a very sensible combination for this task, which is a positive aspect of the work.

Correctness: Appears to be correct, but not all details of the algorithmic implementation are provided in the manuscript. There is no supplementary material, so the correctness can only be really judged by looking in detail at the source code (which I did not do).

Clarity: Yes, it is overall pretty well written and relatively easy to read. Though some of the experimental details could have been described a little more clearly (for example, integrator time-steps, etc.)

Relation to Prior Work: Yes, it is clearly discussed, though not always entirely justified or it remains unclear why the approach was not tested with respect to other related methods (e.g., SINDY).

Reproducibility: Yes

Additional Feedback: - It would be good to add a definition of what is meant by a black-box approach versus the proposed approach. - The clarity of Sec. 3.2 could be improved. If I understand it correctly the networks only take the position and momentum variables respectively as inputs, form the necessary terms of the basis (for example, the desired powers) and then the parameters of the network simply become the coefficients in this basis. So in a sense the networks are linear in the parameters they estimate. Is this the correct understanding? - A few times, statements appear to be a bit strong. For example, in line 155 it is stated "... do not require specialized GPU hardware once trained. For applications such as weather balloons or spacecraft, this is an extremely helpful feature." Can this really be concluded from the experiments? I appreciate the promise, but a some of these applications likely have more complex structures it would be good to either do related experiments or weaken such statements. - Sec. 4.1: What is the time-step of the used integrator for these experiments? In Sec. 6.1 it states it is 0.1, is this the same in Sec. 5.1? If so are the predictions from 0 to 0.1 only based on one time-step? And does Fig. 2 then show long-term integration results? - line 317: "Due to the incorporation of a symplectic integrator, SSINNs make a number of assumptions about the dynamical system that they are approximating, such as conservation of energy and separability." Are these not assumptions that come from the Hamiltonian formulation rather than from the integrator itself? The symplectic integrator only assures that these properties are preserved, no? - There are a couple of recent works that appear to be relevant. These did not factor into my review score (as they all have either only recently been published at ICLR / AISTATS or are still on arxiv), but I am listing them here as they might be of interest to the authors in case they have not seen them yet: a) "Variational Integrator Networks for Physically Structured Embeddings", AISTATS 2020, Steindor Saemundsson, Alexander Terenin, Katja Hofmann, Marc Deisenroth. b) "Hamiltonian Generative Networks", ICLR 2020, Peter Toth, Danilo J. Rezende, Andrew Jaegle, Sébastien Racanière, Aleksandar Botev, Irina Higgins. c) "NeuPDE: Neural Network Based Ordinary and Partial Differential Equations for Modeling Time-Dependent Data", arXiv, 2019, Yifan Sun, Linan Zhang, and Hayden Schaeffer. After rebuttal: My questions have largely been answered by the rebuttal. The rebuttal states. for example. that 1) SINDY experiments will be included, that 2) experiments regarding Hamiltonians which cannot be directly captured by the chosen basis (an issue also raised by R3) will be included, and 3) that the symplectic integrator is indeed important (and RK4 would not work). These are good points, but I wished these aspects would have already been considered before the submission. If all of this material were included I would certainly be happy to raise my review score slightly, but since this is not how the paper was written (and hence this would now require anticipating what a revised paper may look like) I kept the original review score. I encourage the authors to revise the paper and add this additional material for a future submission.


Review 3

Summary and Contributions: This paper proposes a model, SSINN, that learns the underlying Hamiltonian function of a Hamiltonian system symbolically by leveraging performing symplectic integration of the parameterized system as well as sparse regression.

Strengths: The model is novel to my knowledge as an effort to combine symbolic regression and Hamiltonian learning. The empirical evaluations are performed and stated clearly.

Weaknesses: In all the numerical experiments, the functions that appear in the actual Hamiltonian function are more or less contained in the library of functions to search from - even in the last experiment, cosine can be obtained from sine with a shift. Though it is nontrivial to demonstrate that learning can be done in such cases, it is not completely surprising that SSINN can outperform SRNN in these tasks, as the latter model has no prior knowledge of the search space of functions. It would be interesting to perform experiments where the actual Hamiltonian function can not be expressed exactly as a linear combination of functions in the search space.

Correctness: Yes, they look correct to me.

Clarity: Yes, the paper is easy to understand.

Relation to Prior Work: Yes.

Reproducibility: Yes

Additional Feedback: One other baseline method worthy of comparing against is using SINDy without the Hamiltonian inductive bias, which means just learning it as an ODE system, and perhaps also using a symplectic integrator to unroll the predictions. =====Update===== Thanks for the response, and I found my questions addressed. I am glad the comparison with SINDy is added. Regarding the additional experiment on learning a cosine Hamiltonian with polynomial bases, in which a Taylor expansion of the Hamiltonian is learned, I am not sure if this is necessarily a positive result, because with a finite number of polynomials you don't get a good approximation beyond some finite region. Therefore, I wouldn't change my view on this concern.

[Author Response · NeurIPS 2020]

We thank all reviewers for their valuable comments and suggestions. We will release our codebase, create a supplement with pseudocode for our integrator, and make all requested textual changes such as adjusting overly strong wording, clarifying the state-of-the-art, and including all suggested references. Our submission is the first to use a symplectic bias with sparse identification and was described by reviewers as "very sensible" and "of interest to the NeurIPS community." We believe concerns can be addressed within the review cycle via textual changes and further results.

**Comparison to SINDy (R2, R3)**   We have obtained and will incorporate a comparison to SINDy, along with its experimental details. SINDy may be applied by learning a system of ODEs, constructing a Hamiltonian, and predicting with a symplectic integrator if the Hamiltonian is separable. There were no instances where SINDy outperformed our approach; the SINDy mass-spring Hamiltonian was not even separable. SINDy learned the Henon-Heiles Hamiltonian to $10^{-2}$ precision ($10^{-5}$ for an SSINN) and the coupled oscillator Hamiltonian to $10^{-1}$ precision ($10^{-6}$ for an SSINN).

Henon-Heiles Comparison (Fig. 2)

**Hamiltonians not in the basis (R2, R3)**   We will incorporate an additional result into Section 7 using a polynomial basis to approximate the pendulum Hamiltonian. As the cosine term is not present, this SSINN instead converged to the Taylor polynomial of cosine (with the degree depending on the basis used). We assume that Hamiltonians not included in the basis can usually be well-approximated by large polynomial or trigonometric bases. In these instances, sparsity likely remains beneficial, as Taylor polynomials are generally sparse in a polynomial function space. Although simple, this additional example offers some experimental support.

**Non-(p, q) data (R1, R2)**   Rather than incorporating examples with autoencoders, we chose to instead emphasize other common real-world limitations like small and noisy data; many works since Greydanus, et al. have taken a similar approach. We will add lack of non-(p, q) data to Section 8 (Scope and Limitations), along with relevant citations.

**Architecture clarifications (R1, R2)**   SSINNs are presented in Section 3.2 as a single fully-connected linear layer with no bias. This layer learns the necessary terms of the basis, meaning that the trainable parameters become the coefficients and are linear with respect to each term in the basis. However, the SSINN used with Equation 9 in Section 7 is slightly more complicated, including an additional layer with bias for the trainable parameters inside of the sine function (which are learned non-linearly with respect to the model output). We will improve Section 3.2 and Figure 1 by clarifying that, although a single layer with no bias is used for a polynomial basis, more complicated bases (such as Equation 9) may require additional layers with bias terms.

**Experimental clarifications (R1)**   The novel trajectories referenced in the paper were completely held out from the training code and only used for assessing the performance of already trained models. Hyperparameter tuning was slight, consisting of a small grid-search (LR=$[10^{-2}, 10^{-3}]$, L1=$[10^{-4}, 10^{-3}]$), sometimes followed by additional tweaking of regularization. When referring to convergence to the true governing equation, we generally tried to include the decimal precision of convergence (although we used the right sparsity pattern for Section 6). Training and inference speed is comparable to SRNNs but lags behind SINDy due to substantial overhead from the symplectic integrator. We will clarify these details in the paper. We did not incorporate thresholding but like the idea and will mention it in Section 3.2.

**Integrator clarifications (R2)**   Symplectic integrators do maintain conserved quantities, but small oscillations commonly arise in practice, often from truncation error. These oscillations are more apparent for learned Hamiltonians because their trainable parameters tend to have much longer decimal components than the coefficients in the true Hamiltonians–as a result, they are more strongly affected by truncation error. All experiments used $\Delta t = 0.1$; Figure 2 shows long-term integration results (80 steps). We will clarify this in the final paper and reword line 317.

**Drivers of performance (R2)**   The most important driver of performance increases over SRNNs is the simplicity of the chosen basis (and presumed sparsity). Our SRNNs employed the same symplectic integrator as our SSINNs but were still often outperformed by large margins. That said, RK4-trained SSINN Hamiltonians are heavily degraded due to noise from the less accurate integrator, chaos, and lack of symplectic preservation in training. Henon-Heiles 0.1-step test predictions from an RK4-trained Hamiltonian, even when computed with a symplectic integrator, had an average L1-error 4 orders of magnitude larger than the SI-trained Hamiltonian. We will incorporate this discussion.

**Interpretability (R2)**   Throughout the paper, we refer to interpretability in the sense of being able to discover the true Hamiltonian if it is included in the basis, which is not possible for previous approaches with symplectic biases. For clarity, we will adjust our wording and include a definition of what we mean by black-box approaches.

[Meta-Review · NeurIPS 2020]

All three reviewers participated in the discussion, and found the approach appealing. Two of the reviewers feel it is critical that the content which was added by, or promised in, the author response, makes it into the final camera-ready version of the paper. In particular, in addition to all the clarifications included in the author rebuttal, this includes: * Experiments comparing against SINDY * Experiments on Hamiltonians that cannot be captured by the given basis (e.g. finding Taylor expansions with polynomials) * Whether the symplectic integrator is indeed essential (and whether RK4 would work) Additionally, I found it necessary to check the source code for some details — it would be good to explicitly describe the integrator itself (perhaps in an appendix), and explain how it is used in conjunction with automatic differentiation.